# Antimicrobial Stewardship for Outpatients with Chronic Bone and Joint Infections in the Orthopaedic Clinic of an Academic Tertiary Hospital, South Africa

**DOI:** 10.3390/antibiotics12071142

**Published:** 2023-07-01

**Authors:** Mankoana A. Masetla, Pinky N. Ntuli, Veena Abraham, Brian Godman, Bwalya A. Witika, Steward Mudenda, Phumzile P. Skosana

**Affiliations:** 1Department of Clinical Pharmacy, School of Pharmacy, Sefako Makgatho Health Sciences University, Molotlegi Street, Ga-Rankuwa, Pretoria 0208, South Africa; 201601269@swave.smu.ac.za; 2Department of Pharmacy, Dr. George Mukhari Academic Hospital, Molotlegi Street, Ga-Rankuwa, Pretoria 0208, South Africa; 201117228@swave.smu.ac.za; 3Department of Pharmaceutical Sciences, School of Pharmacy, Sefako Makgatho Health Sciences University, Molotlegi Street, Ga-Rankuwa, Pretoria 0208, South Africa; veena.abraham@smu.ac.za (V.A.); bwalya.witika@smu.ac.za (B.A.W.); 4Department of Public Health Pharmacy and Management, School of Pharmacy, Sefako Makgatho Health Sciences University, Pretoria 0208, South Africa; brian.godman@smu.ac.za; 5Strathclyde Institute of Pharmacy and Biomedical Sciences, University of Strathclyde, Glasgow G4 0RE, UK; 6Centre of Medical and Bio Allied Health Sciences Research, Ajman University, Ajman P.O. Box 346, United Arab Emirates; 7Department of Pharmacy, School of Health Sciences, University of Zambia, Lusaka P.O. Box 50110, Zambia; steward.mudenda@unza.zm

**Keywords:** bone and joint infections, antimicrobial stewardship, antimicrobial resistance, guidelines, South Africa

## Abstract

Bone and joint infections are associated with prolonged hospitalizations, high morbidity and complexity of care. They are difficult to treat, and successful therapy requires organism-specific antimicrobial therapy at high doses for a prolonged duration as recommended in standard treatment guidelines (STGs). Adherence to the treatment plan is equally important, which is enhanced with knowledge of the condition as well as appropriate antibiotics. Consequently, the aim of this study was to provide antimicrobial stewardship (AMS) services to outpatients with chronic bone and joint infections presenting to the orthopaedic clinic at a public South African tertiary hospital. A total of 44 patients participated in this study. Chronic osteomyelitis was diagnosed in 39 (89%) patients and septic arthritis in 5 (11%). The majority (43%) of infections were caused by *Staphylococcus aureus* followed by *Pseudomonas aeruginosa* (14%). Seventy-one antibiotics were prescribed at baseline with rifampicin prescribed the most (39%), followed by ciprofloxacin (23%). The majority (96%) of the antibiotics were not prescribed according to the South African STG; however, interventions were only needed in 31% of prescribed antibiotics (n = 71) since the STG only recommends empiric therapy directed against *Staphylococcus aureus*. Seventy-seven percent of the patients obtained a high self-reported adherence score at baseline. Consequently, there is a need to improve AMS in bone and joint infections to improve future care.

## 1. Introduction

Bone and joint infections are infectious diseases that can arise due to the haematogenous spread of systemic bacteraemia, contiguous spread from adjoining tissue or direct trauma caused by surgery or injury [1,2,3,4]. The most common chronic bone and joint infections include osteomyelitis, infectious/septic arthritis and implant/prosthetic-related infections [5]. These infections pose a serious risk of morbidity, and can result in chronic pain, wounds with discharge, sepsis and even permanent disability [6]. Currently in Africa, there is limited knowledge regarding the prevalence of bone and joint infections amongst adults. However, a reported incidence of 43–200 cases per 100,000 children from low-income countries appreciably contrasts with 1.94–13 cases per 100,000 from high-income countries, potentially reflecting a higher incidence of chronic infections in Sub-Saharan Africa [7]. The incidence of bone and joint infections is higher in developing countries due to principally inadequate treatment, health service inaccessibility and poverty, both in seeking care as well as paying for treatment [8].

The most common causative pathogens of bone and joint infections are *Staphylococcus aureus* (*S. aureus*) (the most prevalent single pathogen in bone and joint infections), *coagulase –negative staphylococci* and *Staphylococcus epidermidis* [2,4,5]. Gram-negative bacilli, anaerobes and fungi, have been noted as other causative organisms [4]. Chronic bone and joint infections can also arise due to surgically implanted devices when bacteria adhere to the implant’s surface or a biofilm is formed at the implantation site [9]. To treat these infections, long-term antibiotic treatment at high doses combined with surgery and removal of the implant is necessary [10]. 

The selection of an antibiotic to effectively eradicate the infection depends on a number of factors. These include antibiotic susceptibility, documented bone and joint tissue penetration, oral bioavailability and cost [5]. In South Africa, the Standard Treatment Guidelines/Essential Medicine List (STG/EML) currently recommends empiric antibiotic therapy for osteomyelitis and septic arthritis directed against *S. aureus* [11]. As a result, rflucloxacillin and cloxacillin are recommended as the mainstay of antibiotic therapy [11], with similar recommendations in the recent WHO AWaRE antibiotics guidance [12]. A summary of treatment for these patients according to the current STG is shown in Table 1.

However, biofilm formation is also a critical aspect to consider when treating bone and joint infections [13]. A biofilm is a multi-structural, diverse community of immobile microorganisms enclosed by an extracellular matrix polysaccharide, which usually develops on non-living surfaces, including sequestrum (dead bone) and implants [13,14]. Biofilms encapsulate groups of organisms that are resistant to antimicrobial agents; consequently, making infections difficult to treat, potentially requiring antibiotics outside of current STG/EML guidelines. 

Considering the long duration of therapy necessary to treat bone and joint infections, patient adherence is another key aspect of successful therapy alongside pertinent antibiotic selection [15]. Adherence to medication is a well-acknowledged challenge in the pharmacotherapeutic management of patients with chronic conditions [16]. To successfully eradicate the infection with antibiotics, patients must adhere to all instructions and follow the exact treatment plan given by healthcare professionals [17]. However, not fully understanding instructions from the prescriber and/or pharmacist could lead to non-adherence, especially since there is a high pill burden in patients with chronic bone and joint infections. Consequently, the necessity of educating the patients on their condition and explaining instructions clearly is crucial [18]. Limited knowledge in patients about their condition and medication can result in poor clinical outcomes and, in this case, potentially increase antimicrobial resistance (AMR) [19]. A study by Saqib et al. (2019) reported that insufficient knowledge of prescribed medicines in patients resulted in their incorrect use, which may lead to treatment failure posing a risk to the health of patients [20]. In addition, insufficient knowledge amongst patients about their condition has been shown to negatively impact on interventions aimed at successfully managing their condition [21]. Patient education and clinical pharmacist consultation for chronic disease co-management has been shown to successfully improve medication adherence [22].

The use of STGs in infectious diseases enhances the appropriate use of antibiotics, which is a key concern in Africa [10,23,24]. STGs provide evidence-based treatment recommendations that standardise treatment approaches [10]. Consequently, they are a key part of antimicrobial stewardship programmes (ASPs) to improve the future use of antibiotics [25,26,27]. Part of ASPs is to ensure compliance with local guidelines, which is the responsibility of all healthcare providers [28]. In this respect, hospital pharmacists can play a crucial role by providing ASP interventions including antibiotic prescription reviews, dose and duration optimisation, as well as the education of patients and other healthcare providers [28,29]. 

In view of concerns with rising rates of AMR across Africa, as well as concerns with adherence to guidelines amongst hospitals across Africa [27,30], including hospitals in South Africa, we wished to provide antimicrobial stewardship (AMS) services to the orthopaedic clinic at a tertiary hospital for outpatients diagnosed with chronic bone and joint infections to improve their care. This included the need to evaluate the use of antibiotics in comparison with the current South African STG, assessing patients’ knowledge of their medication and condition, the importance of adherence as well as potentially educating them on their medical condition and antibiotics where necessary. The findings can subsequently be used to update local guidelines if needed, as well as provide guidance to all key stakeholder groups on the potential role and value of pharmacists conducting AMS activities in an outpatient setting to improve patient outcomes. 

## 2. Results

### 2.1. Patient Demographics

Forty-five (45) patients who met the inclusion criteria were recruited into the study. However, one patient withdrew resulting in a total of 44 patients being enrolled into the study. 

The questionnaire was administered to 44 patients at baseline with scheduled follow-up visits every month, totalling two follow-up visits. However, the number of patients decreased for the follow-up visits with only 36% (n = 16) on follow-up 1 and 11% (n = 5) at follow-up 2. This was due to follow-up dates allocated by the clinic falling after the data collection period, with some patients being hospitalised and some not returning to follow-up visits with unknown reasons. The lack of follow-up in some cases was also due to patients completing their course of antibiotics and, therefore, believing they do not need to return to the clinic. 

Of the total sampled population (n = 44), almost half of patients were between the ages range of 30 to 49 (48%) with the majority being males (86%). Twenty patients (45%) presented with comorbidities, with the most frequently reported being the human immunodeficiency virus (HIV) (n = 12, 46%), hypertension (n = 5, 19%), diabetes mellitus and intravenous (IV) drug users (both at n = 3, 12%). Overall, the majority of the patients were diagnosed with chronic osteomyelitis (n = 39, 89%) with only a limited number of patients (n = 5, 11%) diagnosed with septic arthritis. The detailed demographics of the study sample are depicted in Table 2 below.

### 2.2. Microbiology Results

Bacterial cultures were conducted in 31 (70%) patients. Of these, 24 (77%) had a bacterial pathogen cultured while no growth was observed in 7 (23%) patients. Twenty-eight bacterial pathogens of various species were cultured in the 24 patients, with the confirmed bacterial cultures and bacterial distribution depicted in Figure 1. The most cultured microorganisms were *S. aureus* and *Pseudomonas aeruginosa* (*P. aeruginosa)* accounting for most of the infections at (n = 14) 52% and (n = 4) 15%, respectively. Three patients were infected with more than one bacterial pathogen.

### 2.3. Antibiotic Treatment

A total of 71 antibiotics were prescribed from the 44 patients seen at baseline as some patients were receiving more than one antibiotic (Table 3). The most commonly prescribed antibiotics were rifampicin, ciprofloxacin and cloxacillin. Rifampicin was prescribed in 64% (n = 44) of patients, ciprofloxacin in 36% (n = 44) and cloxacillin in 32% (n = 44). In some cases, these antibiotics were given in combination, with the most common combination being rifampicin with ciprofloxacin (36% of prescriptions) and rifampicin with flucloxacillin (14% of prescriptions). These treatment combinations are correct and are used clinically; however, they are not currently included in the South African STG (Table 1). 

Out of the total number of antibiotics prescribed, 38% were ‘Access’ antibiotics and 62% were ‘Watch’ antibiotics. It was noted that there were no ‘Reserve’ antibiotics prescribed. Table 3 provides the breakdown of prescribed antibiotics per their AWaRe classification. 

Of the 71 antibiotics prescribed, 10% were incorrectly dosed. The most common antibiotic for this were rifampicin (7%), followed by cloxacillin and flucloxacillin at 1.5%, respectively. All the doses were less than the recommended range in the South African STG [11]. Owing to the higher potency of flucloxacillin in comparison to cloxacillin, doses of flucloxacillin were halved when scripts reached the pharmacy as that was the only oral option available in stock. 

Amongst the 16 patients that returned at Follow up 1, treatment was amended in four patients where the physician prescribed flucloxacillin and not cloxacillin to avoid halving the dose when dispensing. Doses prescribed for rifampicin improved at Follow up 1 and 2 as prescribers were prescribing higher doses as recommended in the South African STG [11]. All other doses were correct for both visits.

### 2.4. Adherence 

More than half (n = 39; 89%) of the patients had a score of less than 23 points according to the ARMS scale and were considered as the high-adherent group. The rest obtained scores between 23 and 36 points, which categorises them in the medium adherence group (n = 5; 11%). No patients were categorised as low adherence as no patient scored more than 36 points.

### 2.5. Knowledge of the Condition and Medication

At baseline, the majority of patients (57%, n = 25) did not know the name of their condition or their diagnosis, with only 15 patients (34%) reported knowing their condition. Of the 15 that reported knowing their condition, only one patient knew the actual diagnosis of chronic osteomyelitis and three of chronic septic arthritis, respectively. Four patients reported knowing their condition; however, they gave unrelated or incorrect responses. Knowledge of the name of the condition subsequently improved following input with only 19% of the 16 patients reporting not knowing their condition at Follow up 1 and none at Follow up 2 (Table 4).

With regards to the reason for taking their medication, most patients (55%) gave responses consistent with ‘killing the bacteria” whilst 19 (43%) patients reported not knowing at baseline. One patient said the medication was for cleaning the clot (2%). This patient unfortunately did not return for follow-up visits. Eighty-seven percent of patients knew the purpose of their medication at Follow up 1. By Follow up 2, all the patients knew the purpose of their medication (Table 5).

### 2.6. Interventions

A total of 239 interventions were made in this study. These were grouped into three categories: drug treatment interventions, knowledge of the condition and medication, and education of patients on adherence to the medication prescribed (Table 6). The majority of the interventions were for knowledge of the condition and medication (n = 145, 61%), with educating the patient on the name of the condition accounting for most (35%) and common signs and symptoms the least at only 5% (n = 7). Sixty-five interventions (27%) were made in relation to educating the patients on adherence to prescribed antibiotics and the importance of adherence to help resolve their condition. The majority of the drug treatment interventions (n = 29) were for the appropriate antibiotic selection (62%), followed equally by correcting the dose and educating and advising the prescribers (14%). All interventions made to the prescribers were accepted. 

## 3. Discussion

To the best of our knowledge, this is the first study undertaken in a tertiary institution in South Africa that looked at undertaking and evaluating an ASP at an outpatient clinic focusing on bone and joint infections. The most common infection in our study was osteomyelitis. The majority of the patients in our study were males (86%), even when broken down into the diagnosis (chronic osteomyelitis and septic arthritis), the incidence was still higher in males than females; 34:5 and 3:2, respectively. These findings correlate with other studies that observed the incidence of both diagnoses amongst males and females; for chronic osteomyelitis, a ratio of 4:1 was observed and septic arthritis a ratio of 77:47 was observed [31,32]. This could be due to the fact that there is an increase of bone and joint infections due to trauma including traffic accidents and sport injuries in our population, which is more predominant in males as they engage in more risky behaviour [33]. Similarly, a recent study for skin and soft tissue infections in South Africa showed males being more affected [34].

The majority of patients in our study presented with HIV, hypertension, diabetes and as IV drug users for comorbidities. This mirrors recently published studies in a similar setting in South Africa [34]. This is also similar to the findings of Romano et al. (2011), who reported an increased prevalence of bone and joint infections in immunocompromised individuals, as well as those with certain lifestyle habits, including intravenous drug use [35]. South Africa has been reported to have a high prevalence of co-infection of HIV and tuberculosis by the WHO [36], coupled with an increasing prevalence of diabetes and hypertension [37,38]. Consequently, patients with bone and joint infections need to be carefully managed to improve their outcomes.

Microbiological cultures play a crucial role in confirming infections and guiding subsequent antimicrobial treatment. The most prevalent bacterial pathogen in our study was *S. aureus* (43%) followed by *P. aeruginosa* (14%). These findings are similar to those reported in other studies [1,5,39]. However, a recent study by Masters et al. (2022) reported that whilst *S. aureus* was the most common organism identified in these patients, *Pseudomonas* was one of the least common organisms [2]. These differences could be due to geographical differences, as well as different prevalence rates in the community and hospital setting. We will be following this up in future studies. 

A concern was that rifampicin and ciprofloxacin were amongst the most prescribed antibiotics singly and in combination in our study, with flucloxacillin the third most prescribed with the beta-lactams having proven activity against Gram-positive microorganisms, particularly *S. aureus*, which was the most cultured in our study. Whilst this combination is not currently documented in the South African STG (Table 1), this combination has been shown to be effective in published studies, especially in patients with suspected biofilm formation [5,40,41]. Studies documented a clinical cure rate of up to 100% when this combination was used for 3–6 months [5,40,41]. The current South African STG only gives empiric therapy directed at *S. aureus* and not other organisms or biofilm formations (Table 1) [11]. This needs to be addressed, especially with rifampicin being the cornerstone of treatment against biofilm *S. aureus* musculoskeletal infections [10]. However, whilst its effectiveness has been proven across studies [13], care is required in the South African setting as overexposure may lead to resistance [42]; and in a recent study in South Africa, rifampicin was amongst the top five antibiotics prescribed for skin and soft tissue infections [34]. This is a concern as rifampicin is currently used as one of the first-line antimicrobials in the treatment of tuberculosis in South Africa, which has a high burden of drug-resistant tuberculosis (DR-TB) [43]. The efficacy of the floroquinolones has also been proven in bone and joint infections [5,34,35]. However, in South Africa, ciprofloxacin is not used in the treatment of tuberculosis because of its weak efficacy compared to other fluoroquinolones [44,45,46]. This helps explain why adherence to the current South African STG was very low compared to recent point prevalence surveys (PPS) conducted amongst public hospitals in South Africa. In these two PPS studies, compliance with the South African STG was 93.4% and 90.2% of all the prescriptions reviewed [47,48]. The compliance rate in our study was also appreciably lower than the rate of 55.2% compliance to the South African STGs to treat patients with skin and soft tissue infections in a recent study [34] along with other published studies across Africa [49,50]. Rifampicin was also given in combination with flucloxacillin in our study. Using this combination in bone and joint infections, Frippiat et al. (2004) found that at the end of the study all (100%) patients achieved complete infectious remission with a minimum follow-up period of 12 months [40]. This is encouraging.

However, any guidelines produced need to take account of possible shortages of medicines in the public sector necessitating changes to current recommendations. For instance, due to the routine unavailability of oral cloxacillin in the South African public sector, which is dependent on tender supply, flucloxacillin is recommended as the antibiotic of choice where necessary [51]. There is, though, a higher risk of treatment failure with flucloxacillin as high doses are needed for bone penetration in patients with bone and joint infections. After pharmacist interventions, there was improved prescribing of flucloxacillin instead of cloxacillin in our study, which is also encouraging. Similarly, the dispensing of the correct dosage of flucloxacillin was increasingly observed. This shows the importance of involving hospital pharmacists in ASPs as they can help in identifying dosage errors and instigate necessary changes [29,52]. 

Of equal concern is that both rifampicin and ciprofloxacin are antibiotics in the ‘Watch’ group, and it is essential to ensure their rational use to limit AMR in South Africa, which is a growing problem [53]. We have seen variable use of ‘Watch’ antibiotics amongst hospitals in South Africa depending on their location, as well as specific wards [47,48]. This needs to be monitored as antibiotics in the ‘Watch’ group have a higher toxicity and resistance potential [54]. Consequently, appropriate doses and combinations of antibiotics in patients with bone and joint infections must be carefully considered before their use taking into consideration comorbidities including HIV in this population, as well as recommendations in the recently published internationally accepted WHO AWaRe Book [12]. We will be monitoring this in future studies. 

Initially in our study, suboptimal dosages of rifampicin were being prescribed including 150 mg po daily. This contradicts published studies which recommend 600 mg once daily in patients with bone and joint infections [10]. Following successful pharmacist interventions, rifampicin was correctly prescribed at 300 mg twice daily in these patients. This is also in line with other successful ASP interventions undertaken by hospital pharmacists across Africa [29].

Due to the unavailability of oral cloxacillin in South African public sector facilities dependent on tender supply, flucloxacillin is the recommended drug of choice for therapeutic interchange [51]. Flucloxacillin has been proven to achieve higher serum concentrations levels, double that of cloxacillin [55]. As a result of this pharmacokinetic knowledge, for all prescriptions in which cloxacillin was initially prescribed, flucloxacillin was subsequently dispensed with half the dose. This, though, carries a high risk of treatment failure as high doses are needed for bone penetration in bone and joint infections. After pharmacists’ intervention, there was an improved prescribing of flucloxacillin instead of cloxacillin and, similarly, the dispensing of the correct dosage of flucloxacillin was observed. This further shows the importance of the involvement of a pharmacist in AMS initiatives to help in identifying dosage errors and necessary drug changes [28,29].

According to the ARMS scale, the majority (n = 39; 89%) of the patients had a score of less than 23 points; and, consequently, are considered as having high-adherence, which is encouraging. These findings were similar to a study by Zidan et al. (2018)*,* in which the ARMS overall score was 17.4 amongst 307 patients with diabetes [56]. However, we are aware of the reduced patient numbers attending follow-up visits in our study, which may impact on our findings. We will be following this up in future studies since it is important that patients regularly return to the orthopaedic clinic to appropriately treat these chronic bone and joint infections.

At baseline, 57% and 55% of patients indicated not knowing the name of their condition or the purpose of their medication, respectively. With chronic osteomyelitis being the most diagnosed condition, only 2% of the patients knew this at baseline. However, encouragingly by Follow up 1 and 2, their knowledge had appreciably improved alongside knowledge of the condition for purposes of their medication. This indicated an improvement in patients’ knowledge on their condition and the purpose for their medication following pharmacists’ interventions. The importance of patient education was further emphasized in this study, which is similar to other studies [57,58,59].

Outpatient settings will continue to be an important target for AMS interventions [60], and this study has shown possible areas where a pharmacist can play a role. This includes appropriate drug changes and dosing, education of patients and being involved in the overall management of bone and joint infections along with other healthcare professionals (HCPs). It is evident that the initiation of an ASP in this hospital has increased compliance to recommended antibiotics and assisted in reducing inappropriate dosing, similar to other studies [26,27]. The clear need for a pharmacist in bone and joint infections is evident not only in the education of the patients but also of other HCPs involved, as pharmacists play an important role in driving forward ASPs in Africa and globally [28]. Future studies, though, are needed to ensure this service continues and is sustainable to increase better patient outcomes in this important population group, building on the findings of this initial study.

We are aware of a number of limitations in this study. Firstly, this study was only conducted in one tertiary institution focusing on the outpatient population. However, we chose this hospital for this initial study as it is one of the largest tertiary hospitals in South Africa with a designated orthopaedic outpatient clinic. Secondly, the study was performed over only a three-month period using convenience sampling, giving a small sample size. This was a challenge considering that at follow-up visits, not all patients returned, reducing the sample size even further. Thirdly, we are unaware of studies that used the ARMs scale in patients with bone and joint infections. However as stated, this is a validated and reliable tool commonly used in chronic diseases. In addition, the number of patients assessed at follow-up dropped during the study, which may have impacted on our findings. The researchers were only made aware of the different treatment durations during data collection, which made it challenging to investigate this phenomenon as it was not in the original research proposal. This will be looking at this key area in future studies. Lastly, convenience sampling was used, introducing sampling bias as we only included patients who were available and willing to participate in the study. Despite these limitations, we believe our findings are robust providing direction for the future. This includes a key role for hospital pharmacists progressing ASP activities in this key group of patients. 

## 4. Materials and Methods

### 4.1. Study Design, Setting and Population

This was a prospective, interventional quantitative study conducted from May to July 2021 at the orthopaedic clinic and pharmacy of a provincial tertiary and academic hospital in Gauteng, South Africa. This hospital was chosen for this initial study in this area as it is the leading academic tertiary hospital in the public sector, with a bed capacity of 1650. We adopted a similar approach when investigating the management of patients with skin and soft tissue infections in the public healthcare system in South Africa [34]. This is because if major issues are found in patient management in leading hospitals, these issues are likely to occur in secondary and other public hospitals in South Africa. 

Patients, irrespective of gender, visiting an outpatient orthopaedic clinic during the data collection period were approached to be part of the study, in line with other similar studies [61,62] in which a pharmacist was placed in an outpatient department for interventions. On average, the orthopaedic clinic sees a total of 60 patients with different conditions including paediatric patients each month. However, not all these patients will meet the inclusion criteria. Out of these 60, only approximately 12–14 patients presented with osteomyelitis and septic arthritis, including implant-related infections. Using the Raosoft^®^ 2004 calculator (WinNT/200x/XP) with the total of 60 patients seen a month at the clinic, a 95% confidence interval with a 50% proportion and a 5% margin of error, the total sample size was 53 patients per month. However, to avoid bias during selection, all adult outpatients with chronic bone and joint infections who met the inclusion criteria were included in this study.

Patients were included if they were 18 years or older and diagnosed with a chronic bone and/or joint infection, were prescribed antibiotics, presented to the satellite pharmacy that provides services to the orthopaedic clinic during the study and consented to take part in the study. Patients were excluded if they were under the age of 18 with a chronic bone and/or joint infection, visited the orthopaedic clinic but were not on antibiotics or were in-patients with a chronic bone and/or joint infection.

### 4.2. Data Collection Tools

Three data collection tools were used for this study. The Demographics and Clinical Data Form was developed from a data collection tool from a previous study that also researched the role of a hospital pharmacist in an outpatient clinic in a public hospital [59]. It was used in steps one and three of the study to collect patient demographics, which included their age, gender, level of education and occupation, microbiological test findings, diagnosis and diagnosis date, prescribed medication, dispensed medication and details of the pharmacist intervention. 

The “Knowledge of the condition and purpose for medication” questionnaire checklist was formulated based on a previous study [20]. This questionnaire was used in step two of the study. It assessed whether patients knew about their condition by asking about condition-related parameters. This included the name of the condition, common signs and symptoms, causes as well as medication-related parameters, including the purpose of therapy, individual drug purpose, dose/quantity, frequency, route, side-effects, storage, duration of therapy and repeats. 

The Adherence to Refills and Medication Scale is a standardised and validated 12-item questionnaire. It was developed by Kripalani et al. (2009) to evaluate self-reported adherence to taking and refilling medications amongst patients with chronic diseases [63]. The ARMS scale items/questions were constructed for response on a Likert scale with responses of ‘none’, ‘some’, “most”, or ‘all’ of the time, which were given values from 1 to 4, making the minimum achievable score 12 and highest achievable score 48. A low score indicated better adherence and a high score indicated poor adherence. For quantitative purposes, the score set was divided into three subsets to be able to differentiate between low, medium and high adherence. As a result, a score of 12–22 reflected high adherence, 23–36 medium adherence, and 37–48 low adherence. There were no studies that were published using this scale in chronic bone and joint infections based on the literature review; however, there have been similar studies with this scale, researching other chronic conditions including diabetes and HIV [64,65].

### 4.3. Data Collection Procedure 

Data was collected by the principal investigator (MM) in three steps. The first step involved screening of patient files to identify adult outpatients with chronic bone and/or joint infections. Consent was sought from patients who subsequently met the inclusion criteria. Following this, consenting patients were individually taken to a secluded area where an Adherence to Refills and Medications Scale (ARMS) questionnaire (Appendix A) [63], as well as a Knowledge of the Condition and Medication checklist (Appendix A) [20] were administered. The findings were used to perform a baseline assessment of the patients’ knowledge of their condition, their medication and the importance of adherence, and identify areas for intervention to improve their future care. 

In the second step, the medical records of selected files were reviewed at the pharmacy to obtain demographic and clinical data using the demographics and clinical data form (Appendix A) [59]. Prescriptions were reviewed against the South African STG, as well as available microbiology sensitivity results [11]. Prescribers were contacted regarding any necessary interventions in relation to AMS, and these interventions were subsequently documented. Where culture results were unavailable, the STG-recommended antibiotic regimen was used. However, where the culture results were present, the correct organism-specific antibiotic was dispensed and during counselling, patients were educated on their condition, medication and the importance of adherence (Appendix A). Although the treatment for bone and joint infections is for prolonged periods, medication was dispensed for one month and patients were to be reviewed and asked to come back for follow up after each month. For step three, the patients came on their follow-up dates given by the clinic and the same data collection tools from baseline (Appendix A) were used to assess whether patients better understood their condition and medication, and if they were adhering to their medication. Re-emphasis on the knowledge of their condition, medication and importance of adherence was also made at this time (Appendix A). Both step one and two were performed on the same day, with step three undertaken at follow-up visits/dates. COVID-19 safety protocols were followed throughout the data collection period (Appendix A).

## 5. Conclusions

There were concerns that the South African STG for bone and joint infections was not followed. However, the current STG only covers empiric treatment against *S. aureus*. There were also concerns of appreciable prescribing of rifampicin and ciprofloxacin, both ‘Watch’ antibiotics. After the pharmacist intervention, antibiotic prescribing significantly improved. Overall, the current South African STG for bone and joint infections needs to be improved to take account of the different organisms seen and the potential for shortages.

The patients had little knowledge on their condition, medication and importance of adherence at the start of the study. However, after pharmacist education, these aspects significantly improved. Overall, our pilot study proves the merit of including pharmacists as part of multidisciplinary teams managing patients with chronic bone and joint infections to improve their care and reduce the potential for AMR. We will be following this up in future studies. 

## Figures and Tables

**Figure 1 antibiotics-12-01142-f001:**
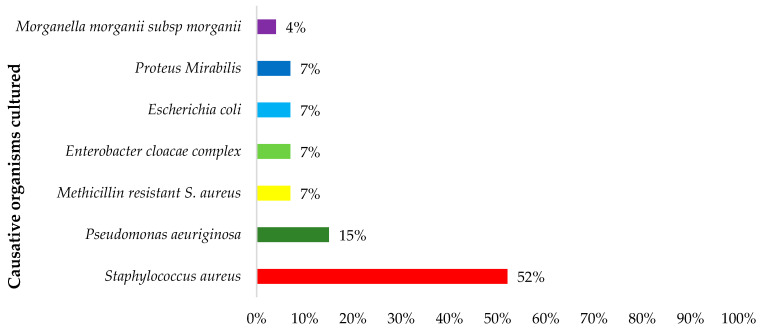
Percentage of causative organisms amongst 44 patients at baseline (n = 27).

**Table 1 antibiotics-12-01142-t001:** South African STG recommendations for osteomyelitis and septic arthritis (adapted from Ref. [11]).

Medication	Route of Administration	Dose	Frequency and Duration
Cloxacillin	IV	2 g	6 hourly for 4 weeks
OR			
Cefazolin	IV	2 g	8 hourly for 4 weeks
After 2 weeks of IV therapy, a change to oral therapy may be considered in patients with a good clinical response.			
Flucloxacillin	oral	1 g	6 hourly to complete the 4 weeks’ treatment
Severe penicillin allergy			
Clindamycin	IV	600 mg	8 hourly for 4 weeks
After 2 weeks of IV therapy, a change to oral therapy may be considered in patients with a good clinical response.			
Clindamycin	oral	450 mg	8 hourly to complete the 4 weeks’ treatment

IV = Intravenous.

**Table 2 antibiotics-12-01142-t002:** Patient demographics.

	Demographics (n = 44)	n (%)
Age groups	18–29	10 (23)
	30–49	21 (48)
	50–69	12 (27)
	70 and above	1 (2)
Gender	Male	38 (87)
	Female	6 (14)
Level of education	Primary school	3 (7)
	High school	33 (75)
	University/college	7 (16)
	No formal education	1 (2)
Comorbidities	HIV positive	12 (46)
	Hypertension	5 (19)
	Diabetes mellitus	3 (12)
	IV drug use	3 (12)
	Epilepsy	1 (4)
	Asthma	1 (4)
	Hypercholesterolemia	1 (4)
Microbiology tests	Conducted	31 (70)
	Not conducted	13 (30)
Diagnosis	Osteomyelitis	39 (89)
	Septic arthritis	5 (11)

HIV = Human immunodeficiency virus; IV = Intravenous.

**Table 3 antibiotics-12-01142-t003:** Antibiotics prescribed at baseline according to AWaRe classification.

AWaRe Classification	Antibiotic	n (%)
Access	Cloxacillin	14 (20)
	Flucloxacillin	11 (15)
	Amoxicillin/clavulanic acid	2 (3)
Watch	Rifampicin	28 (39)
	Ciprofloxacin	16 (23)
Total	71	100%

NB: ‘Access’ and ‘Watch’ are defined in the AWaRe book [12].

**Table 4 antibiotics-12-01142-t004:** Patients’ response to “What is the name of your condition/diagnosis?” at baseline, Follow up 1 and Follow up 2.

	Baseline	Follow up 1	Follow up 2
Number of patients at each visit	44	16	5
‘Don’t know’	25 (57%)	3 (19%)	0
Bone infection	11(25%)	5 (31%)	0
Chronic osteomyelitis	1 (2%)	6 (38%)	4 (80%)
Chronic septic arthritis	3 (7%)	2 (13%)	1 (20%)
Other	4 (9%)	0	0

**Table 5 antibiotics-12-01142-t005:** Patients’ response to “What is the purpose of your medication?” at baseline Follow up 1 and Follow up 2.

	Baseline	Follow up 1	Follow up 2
Total number of patients at each visit	44	16	5
‘Don’t know’	19 (43%)	2 (13%)	0
Kills the bacteria	24 (55%)	14 (87%)	5 (100%)
Other	1 (2%)	0	0

**Table 6 antibiotics-12-01142-t006:** Interventions made during the study.

Categories of Interventions	Intervention	n (%)
Knowledge of condition and medication (n = 145)	Name of the condition	51 (35%)
	Common signs and symptoms	7 (5%)
	Causes of their condition	10 (7%)
	Purpose of therapy	46 (32%)
	Side effects of their medication	12 (8%)
	Duration of treatment	17 (12%)
Adherence (n = 65)	Educating patients on adherence	65 (100)
Drug treatment (n = 29)	Drug change	18 (62%)
	Correcting dose	4 (14%)
	Educating prescribers	4 (14%)
	Correcting frequency	3 (10%)

## Data Availability

Research data can be obtained on reasonable request from the corresponding authors.

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
