# Peer review of "Antimicrobial Stewardship for Outpatients with Chronic Bone and Joint Infections in the Orthopaedic Clinic of an Academic Tertiary Hospital, South Africa"

_antibiotics, 2023, doi:10.3390/antibiotics12071142_

Round 1
Reviewer 1 Report
Thank you for your submission.
I feel that this study is trying to mix multiple items together. The study aim was listed on line 92-93 as "..to evaluate prescribing patterns amongst patients with bone and joint infections starting with tertiary hospitals."
Based on the results listed, I feel that the object of the study really isn't prescribing patterns, but maybe “evaluating health literacy and opportunities for pharmacist-initiated antimicrobial stewardship interventions in patients with chronic bone and/or joint infections.”
Or - Describing clinical and health literacy interventions in patients with chronic bone and joint infections.
Other areas to consider that need clarifying.
Line 61-63. "There are proposals to use nanotechnology...." I am not of the purpose of this sentence. It is not further developed in the discussion. I suggest deleting this sentence.
Line 105-106. follow-up 1 is 1 month later. How long is follow-up 2 from the previous visit. Was the reason for low turnout in follow-up #2 due to patients completing their therapy?
Line 177. Table 4, The row of Other responses at baseline is missing (%)
Line 121. How were the microbiology cultures obtained? was it from surgery, deep tissue, or superficial cultures? If a significant number are from superficial cultures, you will need to add to your limitations the issue of contamination of non-pathogenic organisms.
Line 385. Maybe update the sentence to include, "The findings were used to perform a baseline assessment of the patients’ knowledge of their condition, their medication, the importance of adherence, and identify areas for intervention to improve patient care."
In the limitations section. You should include the small sample size, small and dwindling number of responses to the questionnaire from table 3 and 4. The other item to mention in the study limitation is sampling bias. The study has sampled patients based on convenience.
no detectable issues
Author Response
Reviewer 1
Open Review
( ) I would not like to sign my review report
(x) I would like to sign my review report
Quality of English Language
( ) I am not qualified to assess the quality of English in this paper
( ) English very difficult to understand/incomprehensible
( ) Extensive editing of English language required
( ) Moderate editing of English language required
(x) Minor editing of English language required
( ) English language fine. No issues detected
Author comment: Thank you for this. We have now updated the English with the help of one of the co-authors who is a native English speaker with over 500 publications in peer-reviewed Journals. We trust this is now OK
|
Yes |
Can be improved |
Must be improved |
Not applicable |
|
|
Does the introduction provide sufficient background and include all relevant references? |
( ) |
(x) |
( ) |
( ) |
|
Are all the cited references relevant to the research? |
(x) |
( ) |
( ) |
( ) |
|
Is the research design appropriate? |
(x) |
( ) |
( ) |
( ) |
|
Are the methods adequately described? |
(x) |
( ) |
( ) |
( ) |
|
Are the results clearly presented? |
( ) |
(x) |
( ) |
( ) |
|
Are the conclusions supported by the results? |
( ) |
( ) |
( ) |
( ) |
Comments and Suggestions for Authors Thank you for your submission.
1) I feel that this study is trying to mix multiple items together. The study aim was listed on line 92-93 as "..to evaluate prescribing patterns amongst patients with bone and joint infections starting with tertiary hospitals."
Author comments: Thank you for this. We have now updated the Introduction to clarify the aim of the study. We trust this is now acceptable.
2) Based on the results listed, I feel that the object of the study really isn't prescribing patterns, but maybe “evaluating health literacy and opportunities for pharmacist-initiated antimicrobial stewardship interventions in patients with chronic bone and/or joint infections.” Or - Describing clinical and health literacy interventions in patients with chronic bone and joint infections.
Author comments: Thank you for this comment and recommendations. We have now added information to clarify what the study intended to do, which was to provide antimicrobial stewardship services to the orthopaedic clinic at a tertiary hospital for outpatients diagnosed with chronic bone and joint infections by:
- reviewing the prescriptions in comparison to the STG and making interventions where necessary in consultation with the prescribers
- assessing the patient’s knowledge of their medication and condition, and their knowledge on the importance of adherence
- as well as potentially educating patients on their medical condition and antibiotics where necessary.
We trust this is now OK.
Other areas to consider that need clarifying.
3) Line 61-63. "There are proposals to use nanotechnology...." I am not of the purpose of this sentence. It is not further developed in the discussion. I suggest deleting this sentence.
Author comments: Thank you. We have now deleted this sentence.
4) Line 105-106. follow-up 1 is 1 month later. How long is follow-up 2 from the previous visit. Was the reason for low turnout in follow-up #2 due to patients completing their therapy?
Author comments: Thank you for this. We rephrased the sentence, to make clear that follow up visits were scheduled after every 4 weeks (monthly) by the clinic and not the researcher. We are not sure of the reasons for the fall off in follow-up. This may be for a variety of reasons (as stated) including some patients were allocated follow up dates after the data collection period. The low turnout by the second follow-up was not due to completing their antibiotic course and therefore believing no need for a follow-up and revised the paper accordingly.
5) Line 177. Table 4, The row of Other responses at baseline is missing (%)
Author comments: Thank you. Percentage has been added.
6) Line 121. How were the microbiology cultures obtained? was it from surgery, deep tissue, or superficial cultures? If a significant number are from superficial cultures, you will need to add to your limitations the issue of contamination of non-pathogenic organisms.
Author comments: Thank you for this. However, only information recorded in patient files was collected and reported. More detailed information regarding how the microbiology cultures were obtained was not collected as this fell outside the scope of our study. We acknowledge this limitation and will take it into consideration for future studies.
7) Line 385. Maybe update the sentence to include, "The findings were used to perform a baseline assessment of the patients’ knowledge of their condition, their medication, the importance of adherence, and identify areas for intervention to improve patient care."
Author comments: Thank you. We have updated the sentence as recommended.
8) In the limitations section. You should include the small sample size, small and dwindling number of responses to the questionnaire from table 3 and 4. The other item to mention in the study limitation is sampling bias. The study has sampled patients based on convenience.
Author comments: Thank you and indeed. We have included these in the limitations section.
Comments on the Quality of English Language - no detectable issues
Author comments: Thank you for this – appreciated.
Reviewer 2 Report
The present study is a prospective study conducted in a tertiary hospital from South Africa on a small cohort of 44 patients. It evaluates the ATB prescription patterns for BJIs, and the effects of antimicrobial stewardship (interventions + therapeutical education) on such patterns.
To my mind, the main limit of the study is the small number of patients included initially, and more importantly the small numbers at follow-up visits (follow-up #1, 36% of patients; follow-up #2, 11% of patients). However, since the control of antibiotic use is a major issue in South Africa, and since studies addressing the subject remain scarce, works such as the present one should be encouraged.
Some numbers and percentages should be revised or more clearly presented to ease the reading (see comments below).
-Line 392. “correct medication” is rather vague, does it refer to medications active against documented microorganism(s), or to medications appearing in the guidelines? Were the ATB changed if they differed from the guidelines while being intrinsically active against the documented microorganisms responsible for infection (when microbiological data were available)?
-Line 400. When was the follow-up visit? During treatment? At the end of treatment? If it was only during treatment, how did the authors check that treatment duration was respected by outpatients?
-Line 102. The authors should also mention (in addition to the number of patients enrolled presented here): (i) the total number of orthopedic outpatients during the study period; (ii) the total number of patients who met inclusion criteria. Moreover, the study period should also be clearly stated in the method part (I see 3 months at line 318, in which year?).
- If the clinic sees 60 patients a month (line 336), it makes 180 patients in 3 months (=the study period). Why were only 44 patients included?
-line 106. How many follow-up visits were initially scheduled (it should appear in Fig S1)? I understand the first follow-up was one month later, when was the second follow-up?
-line 108-114. Please revise the numbers and percentages here: I do not understand how, in the same sentence, n=44 may correspond to 48% and 86%, respectively. Moreover, n=44 out of 44 patients included should correspond to 100%?
-Fig 1. Microbiology data are very interesting. Maybe It would have been interesting to know what ATB received these n=27 patients at baseline (and after pharmacist intervention). Maybe as supplementary data?
- Fig 1. Pseudomonas aeuriginosa => Pseudomonas aeruginosa
-Line 131-139. There again, I do not understand the numbers and percentages. How could there be 71 prescriptions of a single ATB (e.g. rifampicin) when there were only 44 patients? Why do percentages vary for a same number (n=71 corresponds to 39%, 23%, and 20% for rifampicin, ciprofloxacin, and cloxacillin, respectively?). Please explain. Same remark at line 170: 55% of 44 patients = 24 patients, not 44.
-The high amount of numbers an Tables make the study very difficult to follow for the common reader. Reading the “2.3. Antibiotic treatment” part, I thought that ATB treatment was changed in seven (line 142) + four (line 147) = 11 patients, while at line 187/Table 5, I see that drugs were changed for 29 patients. Moreover, it would be interesting to know the reason behind the major changes (i.e. drug changes, and dose/frequency changes); was it to stick to South African guidelines? Was it to adapt to bacterial cultures and/or antimicrobial susceptibility testing when available?
-Table 5. How some numbers may be superior to the total number of patients (n=44)?
-Treatment durations are never mentioned throughout the study, why was it not evaluated?
-Maybe a simple reminder of South Africa guidelines in Bone and joint infections should be shown since they are often referred to by the authors (drug? Dose? Duration? Combinations of drugs?)
-Line 229.I fully agree with this part of the discussion (lines229-254). The fact that rifampicin is highly prescribed may be seen as a concern (it could favor the emergence of resistance in TB), but also as something encouraging considering its antimicrobial activity in BJIs due to S. aureus, especially when orthopedic hardware is involved. The authors should add the same remark for fluoroquinolones, since ciprofloxacin seems highly used here: fluoroquinolones are excellent ATBs against BJIs due to S. aureus and Gram-negative bacteria (especially osteitis due to P. aeruginosa), but its high use may increase resistance to fluoroquinolones in TB.
-Did the authors obtain the reason of dosage misuse when chatting with the practitioners? Was it because they did not know the right dose? Was it on purpose to lower the exposition to rifampicin (for example)? Any other reason?
-Maybe another questionnaire could have been used to assess the knowledge and the following of guidelines amongst practitioners? In a further study?
Author Response
Reviewer 2
Open Review
( ) I would not like to sign my review report
(x) I would like to sign my review report
Quality of English Language
( ) I am not qualified to assess the quality of English in this paper
( ) English very difficult to understand/incomprehensible
( ) Extensive editing of English language required
( ) Moderate editing of English language required
( ) Minor editing of English language required
(x) English language fine. No issues detected
Author comment: Thank you for this.
|
Yes |
Can be improved |
Must be improved |
Not applicable |
|
|
Does the introduction provide sufficient background and include all relevant references? |
(x) |
( ) |
( ) |
( ) |
|
Are all the cited references relevant to the research? |
(x) |
( ) |
( ) |
( ) |
|
Is the research design appropriate? |
(x) |
( ) |
( ) |
( ) |
|
Are the methods adequately described? |
( ) |
(x) |
( ) |
( ) |
|
Are the results clearly presented? |
( ) |
( ) |
(x) |
( ) |
|
Are the conclusions supported by the results? |
(x) |
( ) |
( ) |
( ) |
Comments and Suggestions for Authors
1) The present study is a prospective study conducted in a tertiary hospital from South Africa on a small cohort of 44 patients. It evaluates the ATB prescription patterns for BJIs, and the effects of antimicrobial stewardship (interventions + therapeutical education) on such patterns.
Author comments: Thank you for this summary.
2) To my mind, the main limit of the study is the small number of patients included initially, and more importantly the small numbers at follow-up visits (follow-up #1, 36% of patients; follow-up #2, 11% of patients). However, since the control of antibiotic use is a major issue in South Africa, and since studies addressing the subject remain scarce, works such as the present one should be encouraged.
Some numbers and percentages should be revised or more clearly presented to ease the reading (see comments below).
Author comment: Thank you for this. We agree that the numbers are small and have acknowledged this as a limitation in the paper. We are planning, as mentioned, further representative studies with higher number of patients. Thank you also for your comments. We hope we have adequately addressed these.
3) -Line 392. “correct medication” is rather vague, does it refer to medications active against documented microorganism(s), or to medications appearing in the guidelines? Were the ATB changed if they differed from the guidelines while being intrinsically active against the documented microorganisms responsible for infection (when microbiological data were available)?
Author comments: Thank you for the comment. “Correct” in this context means if patients did not have any microbiology results available, then the South African STG recommended antibiotics would be dispensed. Where the microbiology results were available, they would be placed on the antimicrobial that the organism cultured is sensitive to. The sentence has been revised to make this clearer to readers and hope this is now OK.
4) -Line 400. When was the follow-up visit? During treatment? At the end of treatment? If it was only during treatment, how did the authors check that treatment duration was respected by outpatients?
Author comments: Thank you for your comment. We have added more information for clarity, i.e. Although the treatment for bone and joint infections is for prolonged periods, medication was dispensed for one month and patients were to be reviewed and come for follow up after each month. At follow up visits, the pharmacist would expect that the medication should be finished if they were adherent. Hope this is now clear.
5) -Line 102. The authors should also mention (in addition to the number of patients enrolled presented here): (i) the total number of orthopedic outpatients during the study period; (ii) the total number of patients who met inclusion criteria. Moreover, the study period should also be clearly stated in the method part (I see 3 months at line 318, in which year?).
Author comments: Thank you for the comment. This information has now been added to the revised paper. Since the study duration was 3 months, we only managed to enlist 44 patients. We have now added the study duration and specified the year for this initial study. We hope this is now OK.
6) - If the clinic sees 60 patients a month (line 336), it makes 180 patients in 3 months (=the study period). Why were only 44 patients included?
Author comments: Thank you for your comment. As clarified above, the 60 patients was not a true reflection of osteomyelitis, septic arthritis and including implant related infections as was required by this study. We have added more information to clarify and hope this is now acceptable.
7) -line 106. How many follow-up visits were initially scheduled (it should appear in Fig S1)? I understand the first follow-up was one month later, when was the second follow-up?
-line 108-114. Please revise the numbers and percentages here: I do not understand how, in the same sentence, n=44 may correspond to 48% and 86%, respectively. Moreover, n=44 out of 44 patients included should correspond to 100%?
Author comments: Thank you for the comment. We have included in the manuscript that there were two follow up visits which were scheduled monthly and the dates were provided by the clinic. We have added this on Fig S1. The principal researcher who was collecting data would ensure the follow up date was written down and the patient was actively looked for on their follow up dates. The two percentages are from the total sampled population (N=44). We apologize for the confusion and have clarified this in the manuscript.
8) -Fig 1. Microbiology data are very interesting. Maybe It would have been interesting to know what ATB received these n=27 patients at baseline (and after pharmacist intervention). Maybe as supplementary data?
Author comments: Thank you for the comment. We have drawn it up as Table S4.
9) - Fig 1. Pseudomonas aeuriginosa => Pseudomonas aeruginosa
Author comments: Thank you picking this up. The spelling error has been corrected.
10) -Line 131-139. There again, I do not understand the numbers and percentages. How could there be 71 prescriptions of a single ATB (e.g. rifampicin) when there were only 44 patients? Why do percentages vary for a same number (n=71 corresponds to 39%, 23%, and 20% for rifampicin, ciprofloxacin, and cloxacillin, respectively?). Please explain. Same remark at line 170: 55% of 44 patients = 24 patients, not 44.
Author comments: Thank you for your comment. The 71 accounts for all antibiotics prescribed for all the 44 patients. Some patients received more than 1 antibiotic. We have tried to make it clearer in the manuscript, and hope this is now OK.
11) -The high amount of numbers an Tables make the study very difficult to follow for the common reader. Reading the “2.3. Antibiotic treatment” part, I thought that ATB treatment was changed in seven (line 142) + four (line 147) = 11 patients, while at line 187/Table 5, I see that drugs were changed for 29 patients. Moreover, it would be interesting to know the reason behind the major changes (i.e. drug changes, and dose/frequency changes); was it to stick to South African guidelines? Was it to adapt to bacterial cultures and/or antimicrobial susceptibility testing when available?
Author comments: Thank you for your comment. Section 2.3 provides more detail about the total antibiotics prescribed for the 44 patients. The ‘seven’ was for antibiotics incorrectly dosed.
The ‘four’ is not antibiotics. It is the number of patients in which treatment was amended at Follow up 1 and the sample at Follow up 1 was 16 patients.
Table 5 is the number of total interventions made whether it is dose, duration, frequency etc. The total number of interventions was 239. For the table we categorised the type of interventions and 29 of them were drug treatment interventions. We have tried to make this clearer by rephrasing some sections and hope it is now understandable.
12) -Table 5. How some numbers may be superior to the total number of patients (n=44)?
Author comments: Thank you for your comment. As explained above, Table 5 is looking at all interventions made throughout the study. For e.g. a patient can have the wrong dose given, and they don’t know their condition and reason for treatment. Therefore, this patient alone will have three interventions. Hope this brings some clarity. Thank you.
13) -Treatment durations are never mentioned throughout the study, why was it not evaluated?
Author comments: Thank you for your comment. It was not known how much of a huge problem the treatment duration is until data collection commenced. This will be assessed in future studies.
14) -Maybe a simple reminder of South Africa guidelines in Bone and joint infections should be shown since they are often referred to by the authors (drug? Dose? Duration? Combinations of drugs?)
Author comments: Thank you for your comment. The treatment recommendations have been added in the manuscript as Table 1 in the introduction. We hope this is now OK.
15) -Line 229.I fully agree with this part of the discussion (lines229-254). The fact that rifampicin is highly prescribed may be seen as a concern (it could favor the emergence of resistance in TB), but also as something encouraging considering its antimicrobial activity in BJIs due to S. aureus, especially when orthopedic hardware is involved. The authors should add the same remark for fluoroquinolones, since ciprofloxacin seems highly used here: fluoroquinolones are excellent ATBs against BJIs due to S. aureus and Gram-negative bacteria (especially osteitis due to P. aeruginosa), but its high use may increase resistance to fluoroquinolones in TB.
Author comments: Thank you for your comment and very true. We have now added this in the manuscript with reference to studies done in South Africa reporting resistance to ciprofloxacin.
16) -Did the authors obtain the reason of dosage misuse when chatting with the practitioners? Was it because they did not know the right dose? Was it on purpose to lower the exposition to rifampicin (for example)? Any other reason?
Author comments: Thank you for your comment. When liaising with the prescribers, some did not know of the conversion potency of flucloxacillin, others thought we had cloxacillin in stock and other reasons. However, this feedback was not documented as it was not part of the data collection but we will take this into consideration in future studies.
17) -Maybe another questionnaire could have been used to assess the knowledge and the following of guidelines amongst practitioners? In a further study?
Author comments: Thank you for your comment. This will definitely be considered for future studies.
Reviewer 3 Report
Dear Sir,
I found this information interesting, but It its necessary to improve the introduction, since your objective was to evaluate prescribing patters amongst patients but the introduction is about antimicrobial prescription with no mention of patient knowledge of their condition.
The small sample sized could impact the finals results.
Best,
Author Response
Reviewer 3
Open Review
( ) I would not like to sign my review report
(x) I would like to sign my review report
Quality of English Language
( ) I am not qualified to assess the quality of English in this paper
( ) English very difficult to understand/incomprehensible
( ) Extensive editing of English language required
( ) Moderate editing of English language required
( ) Minor editing of English language required
(x) English language fine. No issues detected
Author comments: Thank you for this.
|
Yes |
Can be improved |
Must be improved |
Not applicable |
|
|
Does the introduction provide sufficient background and include all relevant references? |
( ) |
( ) |
(x) |
( ) |
|
Are all the cited references relevant to the research? |
( ) |
(x) |
( ) |
( ) |
|
Is the research design appropriate? |
( ) |
(x) |
( ) |
( ) |
|
Are the methods adequately described? |
(x) |
( ) |
( ) |
( ) |
|
Are the results clearly presented? |
(x) |
( ) |
( ) |
( ) |
|
Are the conclusions supported by the results? |
(x) |
( ) |
( ) |
( ) |
Comments and Suggestions for Authors
Dear Sir,
1) I found this information interesting, but It its necessary to improve the introduction, since your objective was to evaluate prescribing patters amongst patients but the introduction is about antimicrobial prescription with no mention of patient knowledge of their condition.
Author comments: Thank you for this comment. We have added more information in the introduction emphasising patient knowledge of their medication and condition. We trust this is now acceptable
2) The small sample sized could impact the finals results.
Author comments: We acknowledge that the sample size was small for this initial study. We have now added this as a limitation. However, we believe despite the small size we have provided guidance to all key stakeholder groups, and will be following this up. We hope this is acceptable.
Round 2
Reviewer 2 Report
The authors corrected the main issues when compared to the first version of the manuscript. They highlighted the small size of the sample as a limit.
The new presentation of numbers and percentages in text/tables is much more reader-friendly. The same work should be made for the abstract, a naïve reader seeing “interventions were only needed in 31% of patients (n=71)” when there is only n=44 patients included will not understand rightly.
The addition of Supp Table S4 is very interesting in showing the routine everyday work of interventions.
Other remarks below:
- Supp table S4. “Coagulase negative Staphylococcus aureus” => I suppose it is “Coagulase- negative Staphylococcus spp” ? Coagulase-negative S. aureus is rather rare.
- Supp table S4, lines 12 and 16: You state “not covered by ATBs”, but the treatment remains the same after intervention?
- line 157-160: I better understand the numbers after the authors’ explanations. It could be interesting to add the percentage of patients receiving each main ATB (flucloxa, cloxa, cipro, rifam) (rather than the percentage of the 71 ATBs given). For instance, rifam is given to 28/44=60% of patients, which is more impactful than the 28/71=40% given in text.
-Maybe a word could be added about the “treatment duration” issue and why it is never mentioned in the study? In the discussion part, or limit part?
Minor editing of English language required
Author Response
Reviewer 2
Open Review
( ) I would not like to sign my review report
(x) I would like to sign my review report
Quality of English Language
( ) I am not qualified to assess the quality of English in this paper
( ) English very difficult to understand/incomprehensible
( ) Extensive editing of English language required
( ) Moderate editing of English language required
(x) Minor editing of English language required
( ) English language fine. No issues detected
Yes Can be improved Must be improved Not applicable
Does the introduction provide sufficient background and include all relevant references?
(x) ( ) ( ) ( )
Are all the cited references relevant to the research?
(x) ( ) ( ) ( )
Is the research design appropriate?
(x) ( ) ( ) ( )
Are the methods adequately described?
(x) ( ) ( ) ( )
Are the results clearly presented?
(x) ( ) ( ) ( )
Are the conclusions supported by the results?
(x) ( ) ( ) ( )
Comments and Suggestions for Authors
1) The authors corrected the main issues when compared to the first version of the manuscript. They highlighted the small size of the sample as a limit.
Author comments: Thank you so much that this is now acceptable.
2) The new presentation of numbers and percentages in text/tables is much more reader-friendly. The same work should be made for the abstract, a naïve reader seeing “interventions were only needed in 31% of patients (n=71)” when there is only n=44 patients included will not understand rightly.
Author comments: Thank you for your comment. We have taken this to consideration and rephrased the sentence. We hope this is now acceptable
3) The addition of Supp Table S4 is very interesting in showing the routine everyday work of interventions.
Author comments: Thank you for this positive comment – appreciated!
Other remarks below:
4) Supp table S4. “Coagulase negative Staphylococcus aureus” => I suppose it is “Coagulase- negative Staphylococcus spp” ? Coagulase-negative S. aureus is rather rare.
Author comments: Thank you for your comment. The organisms were written as they were found according to the microbiology results in the hospital. We hope this is acceptable.
5) Supp table S4, lines 12 and 16: You state “not covered by ATBs”, but the treatment remains the same after intervention?
Author comments: Thank you for your comment. After consultation with the physician, it was stated that no other effective oral antibiotic could be used in these patients. In this case, the prescribers decided to put the patients on antibiotic suppressive therapy (AST) due to their implants. Consequently, these patients were left on their original medicine.
6) line 157-160: I better understand the numbers after the authors’ explanations. It could be interesting to add the percentage of patients receiving each main ATB (flucloxa, cloxa, cipro, rifam) (rather than the percentage of the 71 ATBs given). For instance, rifam is given to 28/44=60% of patients, which is more impactful than the 28/71=40% given in text.
Author comments: Thank you. we have put this in-text and it now reads “The most commonly prescribed antibiotics were rifampicin, ciprofloxacin and cloxacillin. Rifampicin was prescribed in 64% (n=44) of patients, ciprofloxacin in 36% (n=44) and cloxacillin in 32% (n=44).” We hope this is now OK.
7) Maybe a word could be added about the “treatment duration” issue and why it is never mentioned in the study? In the discussion part, or limit part?
Author comments: Thank you for your comment. We have now amended the Discussion part accordingly and hope this is now acceptable.
Comments on the Quality of English Language - Minor editing of English language required
Author comment. Thank you for this. We have now been through the manuscript with the help of one of the co-authors who is a native English speaker and has over 500 publications in peer-reviewed papers to hie name. We hope this is now acceptable.
Reviewer 3 Report
Dear Sir
All the comments were done.
Author Response
Reviewer 3 – Re-Review
Open Review
( ) I would not like to sign my review report
(x) I would like to sign my review report
Quality of English Language
( ) I am not qualified to assess the quality of English in this paper
( ) English very difficult to understand/incomprehensible
( ) Extensive editing of English language required
( ) Moderate editing of English language required
( ) Minor editing of English language required
(x) English language fine. No issues detected
Author comments: Thank you for this – appreciated!
|
Yes |
Can be improved |
Must be improved |
Not applicable |
|
|
Does the introduction provide sufficient background and include all relevant references? |
(x) |
( ) |
( ) |
( ) |
|
Are all the cited references relevant to the research? |
(x) |
( ) |
( ) |
( ) |
|
Is the research design appropriate? |
(x) |
( ) |
( ) |
( ) |
|
Are the methods adequately described? |
(x) |
( ) |
( ) |
( ) |
|
Are the results clearly presented? |
(x) |
( ) |
( ) |
( ) |
|
Are the conclusions supported by the results? |
(x) |
( ) |
( ) |
( ) |
Comments and Suggestions for Authors
Dear Sir - All the comments were done
Author comments: Thank you for this – appreciated!
